# Availability, Price and Affordability of Anticancer Medicines: Evidence from Two Cross-Sectional Surveys in the Jiangsu Province, China

**DOI:** 10.3390/ijerph16193728

**Published:** 2019-10-03

**Authors:** Yulei Zhu, Ying Wang, Xiaoluan Sun, Xin Li

**Affiliations:** 1Department of Health Policy, School of Health Policy and Management, Nanjing Medical University, Nanjing, Jiangsu 211166, China; 2Department of Clinical Pharmacy, School of Pharmacy, Nanjing Medical University, Nanjing, Jiangsu 211166, China; 3Center for Global Health, School of Public Health, Nanjing Medical University, Nanjing, Jiangsu 211166, China

**Keywords:** availability, price, affordability, anticancer medicines, China

## Abstract

Objectives: With the increasing incidence of cancer, poor access to affordable anticancer medicines has been a serious public health problem in China. To help address this issue, we assessed the availability, price and affordability of pharmacotherapy for cancer in public hospitals in the Jiangsu Province, China. Methods: In 2012 and 2016, anticancer medicine availability and price information in the capital and five other cities was collected. A total of six cancer care hospitals, 26 tertiary general hospitals and 28 secondary general hospitals were sampled, using an adaptation of the World Health Organization/Health Action International methodology. Data was collected for the anticancer medicines in stock at the time of the surveys. Prices were expressed as inflation-adjusted median unit prices (MUPs). Medicine was affordable if the overall cost of all the prescribed anticancer medicines was less than 20% of the household’s capacity to pay. We used generalized estimating equations to estimate the significance of differences in availability from 2012 to 2016 and the Wilcoxon rank test to estimate the significance of differences in MUPs. Multivariate logistic regression was computed to measure predictors of affordability. Results: From 2012 to 2016 there was a significant decrease in the mean availability of originator brands (OBs) (from 7.79% to 5.71%, *p* = 0.012) and lowest-priced generics (LPGs) (36.29% to 32.67%, *p* = 0.009). The mean availability of anticancer medicines in secondary general hospitals was significantly lower than the cancer care, as well as in tertiary general hospitals. The MUPs of OBs (difference: −21.29%, *p* < 0.01) and their LPGs (−22.63%, *p* < 0.01) decreased significantly from 2012 to 2016. The OBs (16.67%) of all the anticancer medicines were found to be less affordable than LPGs (34.62% for urban residents and 30.77% for rural residents); their affordability varied among the different income regions. From 2012 to 2016, the proportion of LPGs with low availability and low affordability dropped from 30.77% to 19.23% in urban areas and 34.62% to 26.92% in rural areas, respectively. Generic substitution and medicine covered by basic medical insurance are factors facilitating affordability. Conclusion: There were concerning decreases in the availability of anticancer medicines in 2016 from already low availability in 2012. Anticancer medicines were more affordable for the patients in high-income regions than the patients in low-income regions. Governments should consider using their bargaining power to reduce procurement prices and abolish taxes on anticancer medicines. Policy should focus on the special health insurance plan for low-income patients with cancer. The goal of drug policy should ensure that first-line generic drugs are available for cancer patients and preferentially prescribed.

## 1. Introduction

Cancer is a major health problem responsible for 15% of overall mortality [1]. With the consolidation of the aging of the population, the number of new cases of cancer worldwide is projected to increase to 21.4 million in 2030 [2]. Globally, the high price of cancer therapy poses a challenge to cancer sufferers and governments alike. Despite much debate about how to reduce the cost of cancer, it continues to increase alarmingly [2]. A study by the Association of Oncology Social Work indicated that the financial burden experienced by cancer patients had negative impacts on their recovery; furthermore, 40% of participates reported that the cost of cancer therapy in the past year swallowed up all their savings [3]. Poor availability and the high cost of cancer treatments are great obstacles to access in many low- and middle-income countries (LMICs), where monthly medicine expenditure often exceeds annual income [4]. Furthermore, these high prices have begun to pose problems in high-income countries. For instance, in the UK, the average cost of cancer treatment had grown 10-fold in the 20 years since 1995 [5].

The mortality and morbidity of cancer have increased in China, which makes cancer the biggest health killer since 2010 [6]. According to the data report through the National Central Cancer Registry (NCCR) of China in 2019, on average, more than ten thousand patients were diagnosed with cancer every day, equating to seven patients being diagnosed with cancer per minute. Consequently, the demand for anticancer medicines will also intensify. Shortages of medicines are occurring at an alarming frequency worldwide [7,8,9,10], which severely affects both adult and pediatric patients with cancer [11,12]. The shortages primarily result from a lack of financial incentives for manufacturing low-cost anticancer medicines, and a weak system for the procurement and distribution of medicines [12,13]. The cost is also a limiting factor for access to anticancer medicines [14]. Statistically, the price of anticancer medicines accounts for approximately a quarter of all cancer costs, which has increased 10-fold in the last decade [15]. For patients with cancer, anticancer medicines are the key to gaining progression-free survival even if there is no overall survival [16].

In 2013, an editorial penned by a large group of experts in chronic myeloid leukemia (CML) drew public attention to the high prices of anticancer medicines [17]. The Union for International Cancer Control (UICC) launched an appeal for improving access to anticancer medicines in lower-resource settings in 2014, which aims to improve inequality in access to anticancer medicines [18]. Sixteen anticancer medicines were added to the 19th revision of the WHO Essential Medicine List in 2015, including three costly medicines [19]. In China, most cancer treatments are not covered by these schemes while the national basic medical insurance coverage is nearly universal [20,21,22]. As in any other country, the affordability of anticancer medicines is a hitherto unknown and grave challenge for the majority of Chinese patients with cancer. A study projected that the financial burden of cancer in China may continue to rise in the coming decades [6]. In 2014, a Chinese merchant named Yong Lu helped several thousand fellow patients buy a generic leukemia drug, Glivec, from India but was investigated for breaching the law. This caused public concern surrounding the accessibility of anticancer medicines to grow [23]. In an attempt to improve drug access, a series of policy initiatives have been undertaken by the Chinese health authority since 2009, with the aim of a material impact on access to anticancer medicines. For instance, the National Health Commission of the People’s Republic of China (NHC) has issued the ‘zero-markup’ drug reform policy with the aim of reducing patients’ medical burden since 2009. As one of the pilot reform provinces, Jiangsu achieved full promotion in public hospitals by the end of 2015. Although the reform attracted extreme concern, little is known about the changes in access to anticancer medicines after the policy was launched.

Price, availability, and affordability are the main criteria to measure whether patients can purchase medicine at an affordable price. A standardized method was developed by the World Health Organization (WHO) and Health Action International (HAI) and has been used in Malaysia, China, Pakistan, Thailand, Sri Lanka, Ethiopia, India, Malawi, Haiti, etc. Most surveys focused on the essential medicines for adults [24,25,26,27,28] and children [29,30,31,32]. Some others focused on medicines for cardiovascular patients [33,34], antidiabetics [35,36], and orphans [37,38]. A few studies have been conducted to evaluate the cost, availability, and affordability of anticancer medicines [39,40,41], but did not cover China. Muhammad et al. found that the availability of both originator brands (OBs) and lowest-priced generics (LPGs) was higher in private sectors (hospitals and pharmacies) than in public hospitals in Pakistan. For both OBs and LPGs, the high-income patients had higher affordability [39]. In another cross-sectional study of anticancer medicines in Pakistan, LPGs had lower availability but higher affordability. Owing to the high price of biologic medicines, oncologists were unwilling to prescribe them and most of the patients were prescribed non-biologics [40]. A survey conducted by the European Society for Medical Oncology (ESMO) demonstrated that the availability of anticancer medicines was low in LMICs, particularly in Eastern Europe. The affordability of cancer therapy with recent market access was the major influencing factor contributing to the inequity of accessibility [41].

In this study, we adopted the WHO and HAI standard methodology to assess the availability, price, and affordability of anticancer medicines for originator brands (OBs) and lowest-priced generics (LPGs) in 60 public hospitals (six tumor hospitals, 26 tertiary general hospitals, and 28 secondary general hospitals) across the capital and five other cities in the Jiangsu Province, China. The OB product had a unique originator pharmaceutical company, and the LPGs were defined as the same product sold under the generic name with the lowest unit price at each hospital at the time of data collection.

We hypothesized that the ‘zero-markup’ drug reform policy would lead to a decrease in the price of anticancer medicine and a gradual increase in the availability of anticancer medicine in health-care institutions. In particular, the imbalance in regional economic development, such as the urban–rural economic disparity was a major factor affecting the accessibility of anticancer medicines. To our knowledge, this is the first study to measure the regional disparity and temporal trends of access to anticancer medicines in China. It was expected that the findings of this study could provide a reference for decision-makers, which could help to improve the accessibility of anticancer medicines in China.

## 2. Materials and Methods

### 2.1. Study Design and Sample

The studies were undertaken in the Jiangsu Province first in 2012 and then again in 2016. The Jiangsu Province is located in the southeastern coast of China, with a population of over 80 million (in 2017) residing across 13 cities. The total gross domestic product of the province was 8586.98 billion RMB (~1271.81 billion USD) in 2017, ranking second of all provinces in mainland China [42]. Notably, cancer is currently one of the top health concerns for the residents of the Jiangsu Province [43]. Two cross-sectional studies were conducted to make comparisons between 2012 and 2016.

According to the different income levels, Jiangsu is divided into three regions, high, middle and low. In total, 60 hospitals providing oncology services were sampled using a multi-stage stratified random sampling. In the first stage, two cities were selected from each region. At the city level, 10 public sectors were selected from each sampling city and comprised 5 tertiary hospitals and 5 secondary hospitals. Taken together, the first stage of the program covered 6 cities in the Jiangsu Province, two high-income (Nanjing and Wuxi), two middle-income (Yangzhou and Nantong), and two low-income regions (Yancheng and Huaian). In each city, we first chose a local specialized tumor hospital as the main survey sector. After these specialized hospitals, additional general hospitals with a Department of Oncology (tertiary general hospitals and secondary general hospitals) were randomly selected within a 3 h drive of the main survey sector. In total, 6 tumor hospitals, 26 tertiary general hospitals, and 28 secondary general hospitals were included in both studies (Table 1). In China, most of the anticancer medicines, restricted prescription drugs, could not be dispensed in community pharmacies. Therefore, in this study, the retail pharmacy outlets were not included in the sampling frame.

### 2.2. Selection of Survey Medicines

According to the requirements of the WHO/HAI methodology, surveyed medicines were selected based on the local patients’ disease burden and needs. In this study, the anticancer medicines were identified based on the local cancer patients’ disease needs, the WHO essential medicine list, the 2012 national essential medicine list (NEML), feedback from several oncologists experts, and literature reviews. Based on the oncologists’ recommendation and local health statistics, a total of 40 medicines were surveyed in both years, selected from the commonly-used medicines for five malignancies (lung cancer, gastric cancer, esophageal cancer, liver cancer, and colorectal cancer) with a high morbidity and fatality rate in the Jiangsu Province. Of the 40 medicines surveyed, 12 were on the WHO essential medicine list and 10 were on the national essential medicine list (NEML). In total, there were 46 anticancer medicines on the WHO essential medicine list and 2012 NEML. The 18 essential medicines sampled covered nearly 40% of the anticancer medicines on the WHO essential medicine list and NEML. For individual medicine, the availability data was collected for two products: OBs and LPGs. Of the 40 medicines surveyed, 15 were for the treatment of lung cancer, 12 were for the treatment of gastric cancer, 8 were for the treatment of esophageal cancer, 7 were for the treatment of liver cancer and 9 were for the treatment of colorectal cancer. Table 2 lists all the surveyed medicines.

### 2.3. Data Collection

A standardized data collection form was designed and made it convenient and efficient for the well-trained research assistants (RAs) to collect information. All the medicines surveyed were noted on a pre-designed sheet. The items were included in the standardized form as follows: facility information (hospital name, hospital level, survey date); medicine information (medicine in stock in the hospital on the day of data collection, yes or no); dosage, strength, medicine type (OB/LPG), retail price. After obtaining permission from the administrative office of the enrolled hospital, twelve research assistants (RAs) visited the hospitals to finish collecting data on the availability and prices of the anticancer medicines for patients. The patient prices were collected for all anticancer medicines found in the hospital on the day of the survey. As a quality control, the RAs checked the completeness and consistency of the search information at the end of each day. The RAs went back on the same day or the next day to fill in missing data.

### 2.4. Data Analysis

The availability of anticancer medicines was calculated as the proportion of all the survey hospitals where the medicine was found on the day of data collection (WHO/HAI 2008). We compared differences in availability of anticancer medicines surveyed between 2012 and 2016 using generalized estimating equations (GEE), and *p* < 0.05 was used to indicate a significant difference. Moreover, we compared the mean availability between different groups of medicines (OBs vs. LPGs; included in the WHO EML or not; included on the NEML or not), across different types of hospitals (tumor hospital, tertiary general hospital and secondary general hospital), across different income levels, and across years (2012 vs. 2016). The following criteria were used for describing the availability of anticancer medicines [28,35,37]:
Absent: 0% of hospitals—these anticancer medicines were not available in any hospital surveyed;Very low: <30% of hospitals—these anticancer medicines were very difficult to find;Low: 30–49% of hospitals—these anticancer medicines were somewhat difficult to find;Fairly high: 50-80% of hospitals: these anticancer medicines were available in many hospitals surveyed;High: >80% of hospitals—these anticancer medicines had acceptable availability.


Price was reported as the median unit price (MUP) in this study. The unit price was accessed by dividing the package price into the package size, which refers to the price per individual tablet, capsule, vial, or milliliter (for bottles) [44]. We also adjusted the 2016 unit prices to 2012 prices by deflating them by 8.5% [45]. According to the WHO/HAI methodology, the MUP of each medicine surveyed was calculated only if the medicine was available in at least four hospitals during the data collection [46]. We compared changes in the MUPs of different medicine groups between 2012 and 2016 and identified whether the MUP decrease or increase for the anticancer medicines between the years was significant using the Wilcoxon rank test.

Cancer treatment lasts for a long period of time and increases health-care costs. The longer the course, the heavier the financial burden the family may suffer. According to the WHO/HAI methodology, generally, if the cost of a course treatment of a medicine is no more than the lowest wage of one day, it is considered affordable. A study published by Rasha Khatib et al. [33] defined it as; “If a treatment cost per month less than 20% of the household capacity-to-pay then it can be considered as affordable.” And the method has been performed in two studies of anticancer medicine affordability in Pakistan [39,40]. In this study, this concept was modified and the affordability was measured for each medicine by low, middle, and high-income class of patients in urban and rural areas through this formula. In this study, on the premise that each of the therapeutic schemes contains a single medicine, the monthly expense of each medicine was denoted in this equation:

The expense of an anticancer medicine per month = Median unit price (MUP) × Defined daily dose (DDD) × 30 days.

The median unit price refers to the median procurement price per unit dose. Defined daily dose (DDD) is a unit of measure that refers to the average maintenance dosage of an anticancer medicine used for its main indication in adults on a daily basis. The DDD information was obtained from the authoritative medicine specification and clinical guidelines. If a treatment cost, per month, less than 20% of the average household monthly income, we regarded it as sufficiently affordability.

We compared the affordability of each medicine to different types of medicines (OBs vs. LPGs), residence (urban area vs. rural area), and both years. The logistic regression model was applied to estimate the independent associations of variables (survey year, medicine type, hospital type, income level, urban vs. rural residence, medicine covered by BMI or not, medicine’s inclusion in the WHO EML, NEML) with the affordability of medicines (affordable or unaffordable). We also performed a comprehensive analysis of the availability and affordability of the anticancer medicines by means of the four-quadrant diagram. The availability value was shown on the X-axis, while the affordability value of the medicines for patients in urban or rural areas was depicted on the Y-axis.

Data were entered using Microsoft Excel and analyzed using SPSS (version 22.0; SPSS, Inc., Chicago, IL, USA) and Matlab (version 2018; MathWorks, Natick, MA, USA).

### 2.5. Ethics Approval

Ethical approval to conduct this study was obtained from the Nanjing Medical University Ethics Committee (grant number: ethical review 201236).

### 2.6. Patient and Public Involvement

In this study, the patients and the public were not involved in the study design.

## 3. Results

### 3.1. Availability

The availability of the 40 anticancer medicines is presented in Table 3. In terms of individual medicines, 13 LPGs (32.50%) had fairly high (50–80%) or high availability (>80%) in 2012 and 14 (35.00%) in 2016, while the availability of OBs was unsatisfactory (<50%) both years. The highest availability of OBs and LPGs was 46.67% and 85.00% in 2012, respectively, and 38.33% and 83.33% in 2016. On average, the total availability of the anticancer medicines decreased significantly from 2012 to 2016 (5.71% in 2016 vs. 7.79% in 2012 for OBs, *p* = 0.012; 32.67% in 2016 vs. 36.29% in 2012 for LPGs, *p* = 0.009). Between 2012 and 2016, non-WHO essential OBs and LPGs showed a significant decrease in mean availability (*p* = 0.028 and *p* = 0.014, respectively), as did non-national essential OBs and LPGs (*p* = 0.013 and *p* = 0.025, respectively). We found that the mean availability of medicines included on the WHO EML was higher than the non-WHO essential medicines. Likewise, the mean availability of medicines included on the NEML was higher than the non-national essential medicines.

The inter-sector comparison of the mean availability of medicines is shown in Table 4. Across hospital types, the mean availability of anticancer medicines surveyed in tumor and tertiary general hospitals was higher than in secondary general hospitals in both time periods. In tumor hospitals, the mean availability of OBs decreased (10.00% in 2016 vs. 10.42% in 2012) and that of LPGs increased (39.17% in 2016 vs. 35.42% in 2012). Compared with 2012, the availability of LPGs included in the WHO EML in 2016 increased significantly (*p* = 0.008) in tumor hospitals. In tertiary general hospitals, the mean availability in 2016 decreased compared with that of OBs (10.29% vs. 11.63%) and LPGs (40.29% vs. 42.31%) in 2012. In secondary general hospitals, the mean availability of OBs decreased (2.68% in 2016 vs. 3.66% in 2012) and that of LPGs increased (32.59% in 2016 vs. 31.07% in 2012).

As shown in Table 5, the mean availability of anticancer medicines surveyed in low-income regions obviously lagged behind that in high- and middle-income regions. The value of availability varied by regions. In low-income regions, the mean availability of OBs remained unchanged (10.13%) and LPGs showed modest improvement (36.13% in 2016 vs 34.50% in 2012). In particular, the mean availability of OBs included in the WHO EML in 2016 increased significantly compared with that in 2012 (*p* = 0.021). In middle-income regions, the mean availability of medicines decreased (7.63% in 2016 vs. 8.88% in 2012 for OBs; 36.63% in 2016 vs. 39.75% in 2012 for LPGs) and significantly changed at OBs (*p* = 0.041). In low-income regions, there was a significant decrease in the mean availability of OBs (2.38% in 2016 vs. 4.38% in 2012, *p* = 0.035) and an increase in that of LPGs (37.00% in 2016 vs. 34.88% in 2012), but no significant change in LPGs (*p* = 0.292).

### 3.2. Anticancer Medicine Prices

Of these 40 anticancer medicines surveyed, only 12 OBs and 26 LPGs were available in four or more hospitals. The adjusted unit prices of each individual medicine in 2016 decreased noticeably compared with those in 2012. The retail prices for OBs were expensive; Trastuzumab ranked the highest (25299.33 RMB in 2012 and 20129.47 RMB in 2016) and Carboplatin had the lowest price (15.39 RMB in 2012 and 11.09 RMB in 2016). For LPGs, Pemetrexed ranked the highest (5935.15 RMB in 2012 and 4722.32 RMB in 2016) and Carboplatin had the lowest retail price (15.39 RMB in 2012 and 11.09 RMB in 2016). A total of 28.95% and 21.05% of the medicines surveyed had a unit retail price (median) above 500 RMB in 2012 and 2016, respectively.

The comparison of the median unit prices in both years and the adjusted changes are shown in Table 6. For OBs, the MUP of medicines in the WHO EML was lower than those not in the WHO EML; for LPGs it was the opposite. The MUP of all medicines on the NEML was lower than that for those not on the NEML. The total MUP of OBs decreased from 549.05 in 2012 to 429.86 in 2016 (*p* < 0.01). The median range of price reduction was between 20.00% and 27.96%. Except for OBs on the NEML, there were significant reductions in other groups of OBs. The total MUP of the LPGs decreased from 120.79 in 2012 to 91.10 in 2016 (*p* < 0.01). Furthermore, all groups of LPGs experienced statistically significant declines in the median unit prices between 2012 and 2016, with the median rate of price reductions ranging from 20.43% to 28.40%.

### 3.3. Affordability of Anticancer Medicines for Standard Treatment Regimens

In total, for 16 OBs and 26 LPGs, the affordability of standard treatment was calculated and found in more than four facilities. Table 7 presents the affordability of the anticancer medicines analyzed. The OBs (16.67% for all residents) of the anticancer medicines were found to be less affordable than LPGs (34.62% for urban residents and 30.77% for rural residents) in both years. In comparison, the least affordable standard treatment was the OB treatment of Pemetrexed for rural residents with low incomes (787.68% in 2012 and 662.36% in 2016).

The universal health insurance system was established in China in 2008 mainly as a basic medical insurance scheme for urban employees, urban residents and rural residents. Currently, both schemes cover the same anticancer medicines divided into two categories and referred to as ‘Class A’ and ‘Class B’. Class A medicines require 0% out-of-pocket payment by patients. Class B medicines require urban employees to pay 25% and general residents to pay 30% of medicinal expenditure. The affordability of each medicine with different levels of insurance reimbursement is provided in Table 8.

After health insurance reimbursement, the affordability improved significantly. In both years, the proportion of affordable OBs increased to 33.33% and 25% for urban and rural residents respectively. Furthermore, in 2012, the proportion of affordable LPGs increased by 38.46% and 57.69% for urban and rural residents respectively. In 2016, the proportion of affordable LPGs increased by 23.07% and 42.31% for urban and rural residents respectively.

Notably, the affordability of each medicine in 2016 showed improvement for both urban and rural residents with each income level in the Jiangsu Province, compared with that in 2012. Anticancer medicines were more affordable for the patients in the high-income region than the patients in the low-income region. For rural residents, the affordability of each medicine for a one month treatment was worse than for urban residents. Compared with 2012, the difference in affordability in 2016 between the rural and urban residents increased. Most medicines required more than 20% of a household monthly income for a one month treatment. The unaffordability of medicines for urban residents in 2012 varied from 22.05% to 616.67%, and from 20.11% to 799.79% for rural residents; the lack of affordability of medicines for urban residents in 2016 varied from 20.56% to 358.41%, and from 20.75% to 672.53% for rural residents. On the whole, the lack of affordability was unreasonable.

### 3.4. Comprehensive Analysis of Medicine Availability and Affordability

Comprehensive analysis of medicine availability and affordability for urban and rural residents is displayed in Figure 1. Figure 1a can be roughly divided into 2 quadrants. The availability of all the OBs was less than 50%. Quadrant I includes 2 (16.67%) medicines with low availability and high affordability, while quadrant II includes 10 (83.33%) medicines with low availability and low affordability. Figure 1b presents a comprehensive analysis of the availability and affordability of LPGs for urban and rural residents in 2012. There were 4 (15.38%) medicines with high availability and high affordability, 9 (34.62%) medicines with high availability and low affordability, 4 (15.38%) medicines with low availability and high affordability and 9 (34.62%) medicines with low availability and low affordability. As shown in Figure 1c,d, there is a decrease in availability but a slightly more optimistic situation in terms of affordability in 2016. Especially for LPGs, there were more anticancer medicines with high availability and high affordability (38.46% for urban residents and 15.38% for rural residents), and less with low availability and low affordability (19.23% for urban residents and 26.92% for rural residents).

Figure 2 presents the comprehensive analysis of medicine availability and affordability following consideration of insurance reimbursement. For urban residents, the number of OBs with low availability and high affordability increased to 3 (25%) and 4 (33.33%) in 2012 and 2016, respectively. The number of LPGs with high availability and high affordability increased to 9 (34.62%) and 14 (53.85%) in 2012 and 2016, respectively. The number of LPGs with low availability and low affordability decreased to 3 (11.54%) in both years. For rural residents, the number of LPGs with high availability and high affordability increased to 9 (34.62%) and 10 (38.46%) in 2012 and 2016, respectively. The number of LPGs with low availability and low affordability decreased to 4(15.38%) and 3 (11.64%) in 2012 and 2016, respectively.

### 3.5. Factors Associated with Affordability

The collinearity test indicated that there was no multicollinearity relationship between these variables (Variance Inflation Factor range from 1.000 to 1.314). The logistic regression model was used to estimate the independent associations of influencing factors (survey year, medicine type, hospital type, income level, urban vs. rural residence, medicine covered by BMI or not, medicine inclusion in the WHO EML or NEML) with the reporting outcome (affordable/unaffordable). However, “types of hospital” (*p* = 1.000) and “medicine inclusion/exclusion on the NEML” (*p* = 0.449) failed to enter the logistic regression equation to forecast the affordability of medicines. Table 9 indicates that the affordability of the anticancer medicines after insurance reimbursement varied significantly with survey year (*p* < 0.001), medicine variety (*p* < 0.001), urban/rural residency (*p* < 0.001), medicine covered by BMI (*p* < 0.001), and medicine inclusion/exclusion in the WHO EML (*p* < 0.001).

Compared with 2012, patients in the Jiangsu Province in 2016 were 0.53-fold more able to afford these anticancer medicines, urban patients were 0.64-fold more than rural, and patients with high incomes were 0.46-fold more than those with low incomes. The LPG medicines were 5.46-fold more affordable than OBs. The medicines included in the WHO EML were 2.75-fold more affordable than those not included. The medicines covered by BMI were 1.59-fold more affordable than those not covered. Furthermore, the odds ratio indicated that generic substitution and medicine covered by BMI are the main contributors to affordability.

## 4. Discussion

In the current study, we adopted the WHO and HAI standard methodology to estimate the availability, price and affordability of anticancer medicines from the surveyed information in 6 cancer care hospitals, 26 tertiary general hospitals and 28 secondary general hospitals in the Jiangsu Province, China. As far as we know, this is the first study to assess the regional disparity and temporal variation of access to anticancer medicines in China. The Jiangsu Province is located in the eastern region of China, which is in the middle-upper level of development of China. As the first anticancer medicine survey to apply the WHO and HAI methodology to the eastern region of China, the findings of this study provide a comprehensive report on the availability, prices and affordability of anticancer medicines in East China. However, due to differences in economic development between Jiangsu and other provinces, there may be regional differences in the affordability of anticancer medicines. The main findings were as follows:

(1) On average, the overall availability of OBs (100%) and LPGs (67.50% for 2012 and 65.00% for 2016) was low (<50%) in both time periods. There was a significant decrease in the mean availability of anticancer medicines surveyed from 2012 to 2016 (from 7.79% to 5.71% for OBs, 36.29% to 32.67% for LPGs). The current study revealed that the mean availability of anticancer medicines was lower in the secondary general hospitals compared to the cancer care, as well as that in the tertiary general hospitals, which was low in low-income areas compared with middle- and high-income areas. All the hospitals surveyed had a higher mean availability of essential versus non-essential anticancer medicines.

The low availability of anticancer medicines is likely multifactorial, such as inadequate investment in research and development [38], insufficient incentive on maintaining stocks and inefficient procurement systems [47]. Our findings are contrary to the research in Pakistan that showed OBs (52.5%) were more available than LPGs (28.1%) in both public and private facilities [39]. Similar to the decreasing trend for availability in the study of essential medicines [48], we noted the decrease in anticancer medicine availability between both time periods. For public hospitals, the sales of medicines accounted for approximately 40% of their revenue [49,50]. Health service providers made higher profits from more costly pharmaceuticals since the mark-up ratio was prescribed by the government. However, the ‘zero-markup’ policy achieved full promotion in the public hospitals of the Jiangsu Province by the end of 2015, which was likely to reduce the hospitals’ revenue. Therefore, the hospitals were unable to maintain these life-saving medicines due to financial constraints. Compared to private sectors, public hospitals are more likely to face the issue of shortage or unavailability of medicines if they don’t receive timely or sufficient financial support to compensate the loss in medicine revenue, especially secondary hospitals [39].

To a large extent, the cost of medicines contributes to availability in the healthcare institutions in LMICs [51]. The regional disparity of availability is of great concern. Availability of anticancer medicines was lowest in low-income cities, which may be due to a lack of sufficient financial support. As extant literature documented, the regions with high-income levels showed a higher availability of medicines, which might be owed to the gaps in the transport system and economic level among different income regions [52]. The higher availability of essential medicines may be due to the National Essential Medicine System established by the central government of China in 2009, which aims to satisfy the public’s basic demand for health care [35,47].

The government has cut down medical expenses to satisfy public demand, while the subsidy for public hospitals remains stationary [50]. To facilitate the availability of anticancer medicines, the government should provide adequate funding to healthcare institutions and improve the equity of the allocation of health resources. In addition, there should be incentives to maintain the stocks of these lifesaving medicines in public hospitals.

(2) The inflation-adjusted median unit prices of anticancer medicines decreased significantly from 2012 to 2016, and were much higher for OBs than LPGs. This finding was consistent with a study done on the prices of the anticancer medicines in 25 provinces [53].

On the one hand, the decrease in retail price of the anticancer medicines may have been due to the ‘zero-markup’ policy [49]. Prior to year 2015, a 15% markup policy was implemented in all of the public hospitals in the Jiangsu Province. Compared with 2012, the cost of anticancer medicines with a zero-markup was much lower. On the other hand, changes to the acquisition method drove the procurement prices down. The procurement became more efficient by negotiating with wholesalers and manufacturers, since the standardized administration in the public bid for drug purchase of public hospitals was strengthened by the central government in 2014 [47]. The pharmaceutical firms could win bids only if they offered the lowest prices. The ‘zero-markup’ reform in the Jiangsu Province achieved remarkable results, attaining diffusion in the whole province and reducing the price of medicines significantly. In China, before June 2015, the price of medicine was regulated by the government. However, it was reported that price regulations had no effect on pharmaceutical price in China [54]. Drug price regulation was replaced by the market pricing policy in June 2015. In this study, we compared changes in the prices of different anticancer medicine groups between 2012 and 2016. A price increase from 2012 to 2016 was not observed in this study. Therefore, the market pricing policy could not have a negative impact on the prices of anticancer medicines.

Drug discovery and development is a complex and lengthy process. In 1994, the significant harmonization of medicine copyrights across national laws was provided by the WTO Trade-Related Aspects of Intellectual Property Rights agreement (TRIPS). The costly anticancer medicines were protected by product patent rights, which made them expensive monopolies [55]. Generally, the patent of new medicines lasts 20 years, which aims to prompt the pharmaceutical firm to recover the costs of research and development and invest again in new research [56]. In addition, high taxation contributes to the sky-high price of these OBs, which is a cruel joke for cancer sufferers [39]. Future policies are urgently needed to reduce the cost of anticancer medicines by the government in order to save lives due to their high mortality. The government should provide adequate funding to make tax-free anticancer medicines more available for cancer patients [40].

(3) Most of anticancer medicines surveyed are not affordable for patients in the absence of health insurance reimbursement. Although the affordability at three different income levels in 2016 showed improvement for both urban and rural residents compared with that in 2012, it was still unreasonable. The OBs (16.67%) of all the anticancer medicines were found to be less affordable than the LPGs (34.62% for urban residents and 30.77% for rural residents); their affordability varied among different income levels and areas. We found that both urban and rural patients with high incomes could not afford the cost of most of the anticancer medicines surveyed, let alone those with middle and low incomes. From 2012 to 2016, the proportion of LPGs with low availability and low affordability dropped from 30.77% to 19.23% in urban area and 34.62% to 26.92% in rural area, respectively. The advent of universal health insurance led to an improvement in the affordability of anticancer medicines in the Jiangsu Province. From 2012 to 2016, the proportion of LPGs with high availability and high affordability rose from 34.62% to 53.85% in urban areas and 34.62% to 38.46% in rural areas, respectively; the proportion of LPGs with low availability and low affordability dropped from 15.38% to 11.64% in rural areas, respectively. Generic substitution and medicine covered by BMI are the main factors facilitating affordability. Due to the bidding system, the pharmaceutical companies that win bids offer the prices. As a result, there is relatively little difference between the prices of the same medicine with same strength and dosage in different types of hospital. Therefore, “types of hospital” failed to enter the logistic regression equation to forecast the affordability of medicines.

Notably, the increased affordability of anticancer medicines has been observed in the Jiangsu Province. This factor might contribute to improved living standards. Meanwhile, the patient prices of anticancer medicines have dropped. However, most anticancer medicines surveyed were not affordable for patients with cancer. Unlike the medicines treating other chronic diseases, the demand for anticancer medicines is largely insensitive or inelastic to the changes in price, due to the life-threatening nature of cancer. Therefore, cancer patients have to be willing to accept and pay the cost of anticancer medicines regardless of their rising costs [57]. Meanwhile, the high cost of anticancer medicines may lead to more households being impoverished and more premature cancer deaths [58].

Policy should focus on addressing the inequity in anticancer medicine’s affordability between patients with different income levels, and between those in urban and rural areas [35]. Our study also revealed that generic substitution could lessen the cost of anticancer medicines to cancer sufferers. Governments should adjust related policies and regulations to make the approvals of generic anticancer medicines faster and improve their access to the market [59]. The preferential taxation policy and other incentive measures can be given to the non-profit generic producers in order to encourage them to manufacture low-cost high-quality anticancer medicines [51]. The government should ensure that first-line LPGs are dispensed in all the public hospitals that provide services to cancer sufferers, and that they are available for patients and preferentially prescribed [47]. Specifically, due to the implementation of the zero-markup policy, drug markups were basically eliminated. It led to a price decrease from 2012 to 2016 at the patient level. On the other hand, medical costs shifted from medicines to other medical services such as surgery and outpatient consultation. For instance, some empirical studies in the Jiangsu Province demonstrated that the total cost of medical care has increased by about 10% with the decline of drug expense after the implementation of the zero-markup policy. Meanwhile, the outpatient consultation and treatment fees have increased significantly [60]. The individual patient’s out-of-pocket expense has decreased slightly [61,62].

Our study has several limitations. First, only the availability of anticancer medicines in stock on the day of data collection was assessed at each facility, which may not accurately capture the availability of anticancer medicines in these hospitals over time. Second, the private hospitals were not considered in the two cross-sectional studies, and therefore are not reflected in the availability of anticancer medicines in all the hospitals in the Jiangsu Province. Third, the affordability may have been overestimated as other treatment costs exceeding one anticancer medicine at a time were not taken into account. The ‘zero-markup’ reform might result in lower costs of medicines but higher health service fees. Fourth, we measured affordability only at the household level. Future studies should focus on the affordability of anticancer medicines at the individual level. Finally, the new anticancer medications that were introduced between 2013 and 2016 were not included, which could lead to selection bias in the price comparison between 2012 and 2016. Due to the fact that most new drugs had higher price tags, the lack of inclusion of new drugs in 2016 could have led to biased lower prices in 2016.

## 5. Conclusions

This study revealed the unreasonable situation of the availability, price and affordability of anticancer medicines. The total availability of the anticancer medicines was low, as a whole, and significantly decreased from 2012 to 2016. In the Jiangsu Province, the price of anticancer medicines was controlled well after the ‘zero-markup’ reform but remained too high. Regional disparity in delivery and affordability of anticancer medicines was observed. There was more availability and affordability for the patients in high-income regions than the patients in low-income regions. In consequence, targeted policy measures should be undertaken to improve the accessibility of anticancer medicines for cancer sufferers. First, having acknowledged the existence of inequities in availability and affordability, a rise in public expenditure on cancer is indispensable. In consideration of the low availability of anticancer medicines in the Jiangsu Province, addressing the availability in hospitals that provide cancer services is a priority. Second, governments should consider using their bargaining power to reduce prices in the purchase phase and abolish taxes on anticancer medicines. Third, policy should focus on the special health insurance plan for the low-income population of cancer sufferers and lower the out-of-pocket payments for cancer therapy. Finally, the health authorities should adjust related policies to make the approvals of generic anticancer medicines faster and improve their access to the market. The goal of drug policy is to ensure that first-line generic drugs are dispensed in public sectors and are available for cancer patients and preferentially prescribed.

## Figures and Tables

**Figure 1 ijerph-16-03728-f001:**
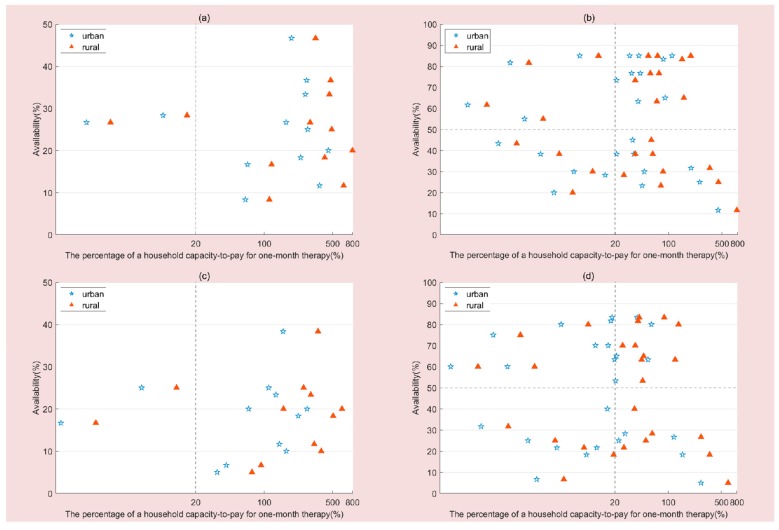
Comprehensive analysis of medicine availability and affordability. (**a**) Comprehensive analysis of OBs availability and affordability in 2012. (**b**) Comprehensive analysis of LPGs availability and affordability in 2012. (**c**) Comprehensive analysis of OBs availability and affordability in 2016. (**d**) Comprehensive analysis of LPGs availability and affordability in 2016.

**Figure 2 ijerph-16-03728-f002:**
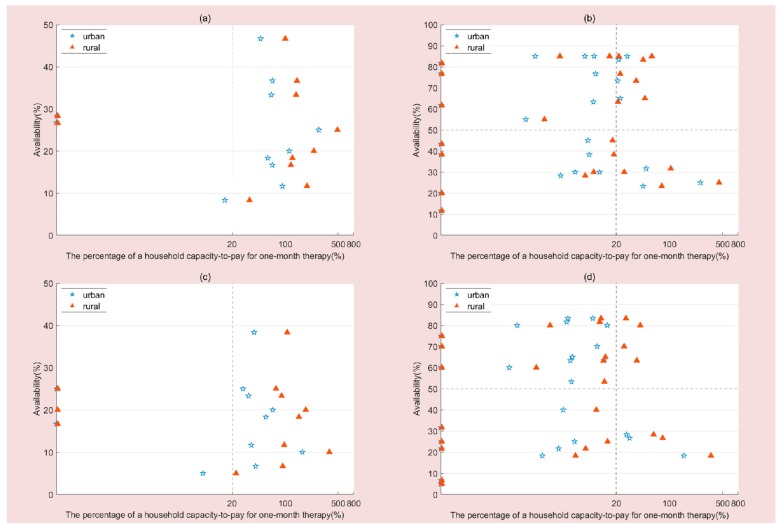
Comprehensive analysis of medicine availability and affordability after insurance reimbursement. (**a**) Comprehensive analysis of OBs availability and affordability in 2012. (**b**) Comprehensive analysis of LPGs availability and affordability in 2012. (**c**) Comprehensive analysis of OBs availability and affordability in 2016. (**d**) Comprehensive analysis of LPGs availability and affordability in 2016.

**Table 1 ijerph-16-03728-t001:** Main characteristics of the 6 sample cities in both surveys.

City	Nanjing (Capital)	Wuxi	Yangzhou	Nantong	Yancheng	Huaian
Economic level	High income	High income	Middle income	Middle income	Low income	Low income
Geographic position	South	South	Center	Center	North	North
No. of hospitals	1 Tumor	1 Tumor	1 Tumor	1 Tumor	1 Tumor	1 Tumor
	4 Tertiary general	5 Tertiary general	5 Tertiary general	4 Tertiary general	4 Tertiary general	4 Tertiary general
	5 Secondary general	4 Secondary general	4 Secondary general	5 Secondary general	5 Secondary general	5 Secondary general

**Table 2 ijerph-16-03728-t002:** List of 40 anticancer medicines surveyed in the Jiangsu Province.

Name	Dosage Form	Strength	Volume	WHO EML	NEML	Insurance Coverage in Jiangsu Province	Main Indication
Bleomycin	Vial	15 IU	1	Yes	No	Class B	Esophageal cancer
Calcium folinate	Vial	0.1 g	1	No	Yes	No	Gastric cancer, colorectal cancer
Capecitabine	TAB-CAP	0.5 g	12	Yes	No	Class B	Colorectal cancer
Carboplatin	Bottle	0.15 g/15 mL	1	Yes	No	Class A	Lung cancer
Carmofur	TAB-CAP	50 mg	24	No	No	Class B	Gastric cancer, esophageal cancer, colorectal cancer
Cetuximab	Bottle	100 mg/50 mL	1	No	No	No	Liver cancer, colorectal cancer
Cisplatin	Vial	20 mg	1	No	Yes	Class A	Lung cancer, liver cancer
Cyclophosphamide	TAB-CAP	50 mg	24	Yes	Yes	Class A	Lung cancer
Cytarabine	Vial	100 mg	1	Yes	Yes	Class A	Lung cancer
Docetaxel	Bottle	20 mg/0.5 mL	1	No	No	Class B	Lung cancer
Doxifluridine	TAB-CAP	0.2 g	50	No	No	Class B	Gastric cancer, colorectal cancer
Doxorubicin	Bottle	20 mg/10 mL	1	No	No	Class A	Lung cancer, gastric cancer, liver cancer
Endostar	Bottle	15 mg/3 mL	1	No	No	No	Lung cancer
Epirubicin	Vial	10 mg	1	No	No	Class B	Lung cancer, gastric cancer, liver cancer
Erlotinib	TAB-CAP	150 mg	7	No	No	No	Lung cancer, liver cancer
Etoposide	Bottle	100 mg/5 mL	1	Yes	Yes	Class A	Lung cancer
Fluorouracil	Vial	0.1 g	1	No	No	Class A	Gastric cancer, esophageal cancer, colorectal cancer
Gefitinib	TAB-CAP	0.25 g	10	No	No	Class B	Lung cancer
Gemcitabine	Vial	0.2 g	1	Yes	No	Class B	Lung cancer
Ifosfamide	Vial	1 g	1	Yes	No	Class B	Lung cancer
Imatinib	TAB-CAP	0.1 g	60	Yes	No	Class B	Esophageal cancer
Irinotecan	Vial	0.1 g	1	No	No	Class B	Lung cancer, esophageal cancer
Lentinan	Vial	1 mg	1	No	No	No	Gastric cancer, liver cancer
Levomisole	TAB-CAP	25 mg	100	No	No	No	Lung cancer
Methotrexate	Vial	0.1 g	1	No	Yes	Class A	Lung cancer, gastric cancer, colorectal cancer
Mitomycin	Vial	2 mg	1	No	Yes	Class A	Lung cancer, gastric cancer
Nedaplatin	Vial	10 mg	1	No	No	Class B	Lung cancer, colorectal cancer
Ondansetron	Bottle	8 mg/4 mL	1	No	No	No	Other adjuvant medicine
Oxaliplatin	Vial	50 mg	1	Yes	Yes	Class B	Colorectal cancer
Paclitaxel	Bottle	30 mg/5 mL	1	No	Yes	Class A	Lung cancer, colorectal cancer
Pemetrexed	Vial	0.5 g	1	No	No	Class B	Lung cancer
Pingyangmycin	Vial	8 mg	1	No	No	Class A	Esophageal cancer
Semustine	TAB-CAP	50 mg	5	No	Yes	Class A	Lung cancer
Sunitinib	TAB-CAP	12.5 mg	28	No	No	No	Liver cancer
Tegafur	Bottle	500 mg/10 ml	1	No	No	Class B	Gastric cancer, colorectal cancer
Tegafur, Gimeracil and Oteracil Porassium Capsules	TAB-CAP	(Tegafur 20 mg, gimeracil 5.8 mg, Oteracil Porassium 19.6 mg)	42	No	No	No	Gastric cancer
Topotecan	TAB-CAP	1 mg	4	No	No	No	Lung cancer
Trastuzumab	Vial	0.44 g	1	Yes	No	No	Gastric cancer
Vindesine	Vial	1 mg	1	No	No	Class B	Lung cancer
Vinorelbine	Bottle	10 mg/mL	1	Yes	No	Class B	Lung cancer

Note: WHO EML—World Health Organization essential medicine list; NEML—National Essential Medicine List; Yes—on the list; No—not on the list; Class A—Class A medicine list of basic medical insurance (BMI) in the Jiangsu Province; Class B—Class B medicine list of BMI.

**Table 3 ijerph-16-03728-t003:** Availability of 40 anticancer medicines in the Jiangsu Province [*n* (%)].

Name		OBs		LPGs	
		2012	2016	2012	2016
Bleomycin		0 (0.00)	0 (0.00)	1 (1.67)	7 (11.67)
Calcium folinate		0 (0.00)	0 (0.00)	18 (30.00)	13 (21.67)
Capecitabine		0 (0.00)	0 (0.00)	51 (85.00)	48 (80.00)
Carboplatin		16 (26.67)	10 (16.67)	N/A	N/A
Carmofur		0 (0.00)	0 (0.00)	17 (28.33)	11 (18.33)
Cetuximab		4 (6.67)	1 (1.67)	N/A	N/A
Cisplatin		0 (0.00)	0 (0.00)	0 (0.00)	11 (18.33)
Cyclophosphamide		0 (0.00)	0 (0.00)	1 (1.67)	5 (8.33)
Cytarabine		17 (28.33)	15 (25.00)	23 (38.33)	15 (25.00)
Docetaxel		22 (36.67)	14 (23.33)	50 (83.33)	50 (83.33)
Doxifluridine		0 (0.00)	0 (0.00)	7 (11.67)	3 (5.00)
Doxorubicin		0 (0.00)	1 (1.67)	7 (11.67)	3 (5.00)
Endostar		0 (0.00)	0 (0.00)	15 (25.00)	11 (18.33)
Epirubicin		0 (0.00)	0 (0.00)	51 (85.00)	50 (83.33)
Erlotinib		15 (25.00)	6 (10.00)	N/A	N/A
Etoposide		0 (0.00)	0 (0.00)	49 (81.67)	45 (75.00)
Fluorouracil		0 (0.00)	0 (0.00)	23 (38.33)	13 (21.67)
Gefitinib		11 (18.33)	7 (11.67)	N/A	0 (0.00)
Gemcitabine		28 (46.67)	15 (25.00)	52 (85.00)	48 (80.00)
Ifosfamide		2 (3.33)	2 (3.33)	27 (45.00)	24 (40.00)
Imatinib		7 (11.67)	11 (18.33)	N/A	0 (0.00)
Irinotecan		0 (0.00)	0 (0.00)	39 (65.00)	38 (63.33)
Lentinan		0 (0.00)	0 (0.00)	44 (73.33)	42 (70.00)
Levomisole		0 (0.00)	0 (0.00)	11 (18.33)	3 (5.00)
Methotrexate		0 (0.00)	0 (0.00)	37 (61.67)	36 (60.00)
Mitomycin		0 (0.00)	0 (0.00)	26 (43.33)	19 (31.67)
Nedaplatin		0 (0.00)	0 (0.00)	46 (76.67)	38 (63.33)
Ondansetron		1 (1.67)	0 (0.00)	14 (23.33)	17 (28.33)
Oxaliplatin		20 (33.33)	23 (38.33)	52 (85.00)	49 (81.67)
Paclitaxel		16 (26.67)	12 (20.00)	46 (76.67)	42 (70.00)
Pemetrexed		12 (20.00)	12 (20.00)	19 (31.67)	16 (26.67)
Pingyangmycin		0 (0.00)	0 (0.00)	22 (36.67)	1 (1.67)
Semustine		0 (0.00)	0 (0.00)	12 (20.00)	4 (6.67)
Sunitinib		1 (1.67)	1 (1.67)	N/A	N/A
Tegafur		0 (0.00)	0 (0.00)	18 (30.00)	15 (25.00)
Tegafur, Gimeracil and Oteracil Porassium Capsules		0 (0.00)	0 (0.00)	33 (55.00)	36 (60.00)
Topotecan		0 (0.00)	0 (0.00)	1 (1.67)	0 (0.00)
Trastuzumab		10 (16.67)	4 (6.67)	N/A	N/A
Vindesine		0 (0.00)	0 (0.00)	23 (38.33)	32 (53.33)
Vinorelbine		5 (8.33)	3 (5.00)	38 (63.33)	39 (65.00)
Avg	Total	4.68 (7.79)	3.43 (5.71)	21.83 (36.29)	19.60 (32.67)
	WHO EM	8.75 (4.72)	6.92 (4.03)	24.50 (43.06)	23.33 (40.14)
	Non-WHO EM	2.93 (4.62)	1.93 (3.54)	20.68 (34.75)	18.00 (30.39)
	NEM	5.30 (6.25)	5.00 (5.64)	26.40 (42.77)	23.90 (39.95)
	Non-NEM	4.47 (4.63)	1.91 (3.55)	20.30 (35.030	17.23 (30.72)

Note: OBs—originator brands; LPGs—lowest-priced generics; Avg—average; N/A—the medicine did not get approval from the department of drug administration in China. WHO EM, WHO essential medicine; Non-WHO EM—non-WHO essential medicine; NEM—national essential medicine; Non-NEM—non-national essential medicine.

**Table 4 ijerph-16-03728-t004:** Mean availability of anticancer medicines in the Jiangsu Province: inter-sector comparison.

		Tumor Hospital (%)		Tertiary General Hospital (%)		Secondary General Hospital (%)	
		2012	2016	2012	2016	2012	2016
**OBs**	All	10.42	10.00	11.63	10.29	3.66	2.68
	WHO EM	18.06	19.44	23.08	19.87	5.95	6.25
	Non-WHO EM	7.14	5.95	6.73	6.18	2.68	1.15
	NEM	8.33	11.67	13.85	13.46	4.29	5.36
	Non-NEM	11.11	9.44	10.90	9.23	3.45	1.79
**LPGs**	All	35.42	39.17	42.31	40.29	31.07	32.59
	WHO EM	36.11	45.83	46.15	48.08	36.90	38.69
	Non-WHO EM	35.12	36.31	40.66	36.95	28.57	29.97
	NEM	38.33	43.33	49.62	49.23	40.00	39.64
	Non-NEM	34.44	37.78	39.87	37.31	28.10	30.24

Note: OBs—originator brands; LPGs—lowest-priced generics; WHO EM—WHO essential medicine; Non-WHO EM—non-WHO essential medicine; NEM—national essential medicine; Non-NEM—non-national essential medicine.

**Table 5 ijerph-16-03728-t005:** Mean availability of anticancer medicines in the Jiangsu Province: inter-region comparison.

		High Income (%)		Middle Income (%)		Low Income (%)	
		2012	2016	2012	2016	2012	2016
**OBs**	All	10.13	10.13	8.88	7.63	4.38	2.38
	WHO EM	20.42	21.25	15.00	13.33	8.33	5.83
	Non-WHO EM	5.71	5.36	6.25	5.18	2.68	0.89
	NEM	12.50	15.00	8.00	10.00	6.00	3.50
	Non-NEM	9.33	8.50	9.17	6.83	3.83	2.00
**LPGs**	All	34.50	36.13	39.75	36.63	34.88	37.00
	WHO EM	33.75	41.25	46.67	45.42	42.08	43.75
	Non-WHO EM	34.82	33.93	36.79	32.86	31.79	34.11
	NEM	41.50	45.50	46.50	42.00	44.00	45.00
	Non-NEM	32.17	33.00	37.50	34.83	31.83	34.33

Note: OBs—originator brands; LPGs—lowest-priced generics; WHO EM—WHO essential medicine; Non-WHO EM—non-WHO essential medicine; NEM—national essential medicine; Non-NEM—non-national essential medicine.

**Table 6 ijerph-16-03728-t006:** Median unit prices of anticancer medicines and the median rate of price changes in 2012 and 2016.

	2012 MUPs (*n*)	2016 MUPs (*n*)	Median Rate of Change (%)
**OBs**			
All	549.05 (12)	429.86 (12)	−21.29 **
WHO EM	406.97 (7)	275.30 (7)	−22.14 *
Non-WHO EM	691.43 (5)	550.13 (5)	−20.00 *
NEM	313.95 (3)	170.74 (3)	−22.14
Non-NEM	586.50 (9)	466.65 (9)	−20.43 **
**LPGs**			
All	120.79 (26)	91.10 (26)	−22.63 **
WHO EM	148.35 (7)	92.30 (7)	−28.40 *
Non-WHO EM	117.97 (19)	89.91 (19)	−21.14 **
NEM	18.69 (8)	14.82 (8)	−20.43 *
Non-NEM	173.31 (18)	114.57 (18)	−24.29 **

Note: OBs—originator brands; LPGs—lowest-priced generics; WHO EM—WHO essential medicine; Non-WHO EM—non-WHO essential medicine; NEM—national essential medicine; Non-NEM—non-national essential medicine; MUP—median unit price (RMB); *n*—number of sectors surveyed; * Wilcoxon test *p* < 0.05, ** *p* < 0.01.

**Table 7 ijerph-16-03728-t007:** Affordability of anticancer medicines in the urban and rural areas in different regions of the Jiangsu Province.

Drug Name	Type	Total (%)		High Income (%)		Middle Income (%)		Low Income (%)	
		2012	2016	2012	2016	2012	2016	2012	2016
**Urban area**									
Calcium folinate	LPGs	5.71	3.45	4.69	2.74	6.09	3.49	7.77	4.51
Capecitabine	LPGs	6.80	3.92	5.59	3.11	7.26	3.96	9.26	5.12
Carboplatin	OBs	1.53	0.84	1.25	0.66	1.63	0.85	2.08	1.09
Carmofur	LPGs	14.69	8.46	12.08	6.72	15.67	8.55	19.99	11.06
Cytarabine	OBs	9.24	5.59	7.60	4.44	9.85	5.65	12.57	7.31
	LPGs	2.08	1.44	1.71	1.14	2.22	1.45	2.83	1.88
Docetaxel	OBs	271.49	131.75	223.27	104.56	289.64	133.13	369.55	172.19
	LPGs	85.23	39.09	70.10	31.02	90.93	39.50	116.02	51.09
Doxorubicin	LPGs	446.18	270.09	366.94	214.36	476.00	272.92	607.34	352.98
Endostar	LPGs	255.50	154.66	210.12	122.75	272.57	156.28	347.78	202.13
Epirubicin	LPGs	30.72	18.36	25.27	14.57	32.78	18.55	41.82	23.99
Erlotinib	OBs	278.16	168.38	228.76	133.64	296.75	170.14	378.63	220.06
Etoposide	LPGs	0.83	0.50	0.68	0.40	0.89	0.51	1.13	0.66
Fluorouracil	LPGs	20.73	11.54	17.05	9.16	22.11	11.66	28.22	15.09
Gefitinib	OBs	235.95	142.83	194.04	113.36	251.72	144.32	321.17	186.66
Gemcitabine	OBs	190.23	111.19	156.44	88.25	202.94	112.35	258.93	145.32
	LPGs	110.73	60.32	91.06	47.87	118.13	60.95	150.72	78.83
Ifosfamide	LPGs	33.62	15.91	27.65	12.63	35.87	16.08	45.76	20.80
Imatinib	OBs	368.27	222.93	302.86	176.92	392.88	225.26	501.28	291.34
Irinotecan	LPGs	89.83	54.38	73.87	43.15	95.83	54.94	122.27	71.06
Lentinan	LPGs	20.67	11.10	17.00	8.81	22.05	11.22	28.14	14.51
Methotrexate	LPGs	0.23	0.14	0.19	0.11	0.25	0.14	0.31	0.18
Mitomycin	LPGs	0.58	0.35	0.47	0.28	0.61	0.35	0.78	0.46
Nedaplatin	LPGs	42.40	19.76	34.87	15.68	45.23	19.96	57.71	25.82
Ondansetron	LPGs	44.90	27.18	36.92	21.57	47.90	27.46	61.11	35.52
Oxaliplatin	OBs	263.59	156.14	216.78	123.92	281.21	157.78	358.80	204.06
	LPGs	40.63	17.68	33.41	14.03	43.34	17.87	55.30	23.11
Paclitaxel	OBs	167.44	69.28	137.70	54.98	178.63	70.01	227.92	90.54
	LPGs	32.65	16.27	26.85	12.91	34.83	16.44	44.44	21.27
Pemetrexed	OBs	453.04	274.24	372.58	217.65	483.32	277.11	616.67	358.41
	LPGs	196.70	119.07	161.77	94.50	209.85	120.32	267.75	155.61
Semustine	LPGs	3.12	1.87	2.57	1.49	3.33	1.89	4.25	2.45
Tegafur	LPGs	47.90	22.40	39.39	17.78	51.10	22.64	65.20	29.28
Tegafur, Gimeracil and Oteracil Porassium Capsules	LPGs	1.28	0.77	1.05	0.61	1.36	0.78	1.74	1.01
Trastuzumab	OBs	67.60	40.92	55.59	32.48	72.12	41.35	92.02	53.48
Vindesine	LPGs	35.09	20.35	28.86	16.15	37.44	20.56	47.76	26.59
Vinorelbine	OBs	64.14	33.01	52.75	26.20	68.43	33.36	87.31	43.14
	LPGs	39.72	20.99	32.66	16.66	42.37	21.21	54.06	27.43
**Rural area**									
Calcium folinate	LPGs	10.04	7.85	7.81	5.02	9.54	7.37	10.07	8.47
Capecitabine	LPGs	11.98	8.91	9.31	5.69	11.38	8.36	12.01	9.61
Carboplatin	OBs	2.69	1.90	2.09	1.22	2.55	1.78	2.69	2.05
Carmofur	LPGs	25.86	19.23	20.11	12.30	24.56	18.06	25.93	20.75
Cytarabine	OBs	16.26	12.71	12.64	8.13	15.45	11.93	16.31	13.71
	LPGs	3.66	3.27	2.84	2.09	3.48	3.07	3.67	3.53
Docetaxel	OBs	477.95	299.46	371.66	191.50	454.04	281.19	479.28	323.09
	LPGs	150.05	88.85	116.69	56.82	142.55	83.42	150.47	95.86
Doxorubicin	LPGs	785.50	613.90	610.82	392.58	746.20	576.44	787.68	662.36
Endostar	LPGs	449.80	351.54	349.77	224.80	427.29	330.08	451.05	379.28
Epirubicin	LPGs	54.09	41.73	42.06	26.68	51.38	39.18	54.24	45.02
Erlotinib	OBs	489.70	382.72	380.80	244.74	465.20	359.37	491.06	412.93
Etoposide	LPGs	1.46	1.14	1.14	0.73	1.39	1.07	1.47	1.23
Fluorouracil	LPGs	36.49	26.24	28.38	16.78	34.67	24.64	36.59	28.31
Gefitinib	OBs	415.38	324.64	323.01	207.60	394.60	304.83	416.54	350.26
Gemcitabine	OBs	334.89	252.73	260.42	161.62	318.13	237.31	335.82	272.68
	LPGs	194.94	137.11	151.59	87.68	185.19	128.74	195.48	147.93
Ifosfamide	LPGs	59.19	36.17	46.03	23.13	56.23	33.96	59.35	39.03
Imatinib	OBs	648.33	506.70	504.15	324.02	615.89	475.77	650.13	546.68
Irinotecan	LPGs	158.14	123.59	122.97	79.03	150.23	116.05	158.58	133.35
Lentinan	LPGs	36.39	25.23	28.30	16.14	34.57	23.69	36.49	27.23
Methotrexate	LPGs	0.41	0.32	0.32	0.20	0.38	0.30	0.41	0.34
Mitomycin	LPGs	1.01	0.79	0.79	0.51	0.96	0.74	1.02	0.85
Nedaplatin	LPGs	74.64	44.90	58.04	28.71	70.91	42.16	74.85	48.45
Ondansetron	LPGs	79.04	61.77	61.46	39.50	75.08	58.00	79.26	66.65
Oxaliplatin	OBs	464.05	354.90	360.85	226.95	440.83	333.24	465.34	382.91
	LPGs	71.53	40.19	55.62	25.70	67.95	37.74	71.73	43.36
Paclitaxel	OBs	294.78	157.47	229.23	100.70	280.03	147.86	295.60	169.90
	LPGs	57.48	36.98	44.70	23.65	54.61	34.73	57.64	39.90
Pemetrexed	OBs	797.57	623.34	620.21	398.61	757.66	585.30	799.79	672.53
	LPGs	346.29	270.64	269.28	173.07	328.97	254.13	347.25	292.00
Semustine	LPGs	5.49	4.26	4.27	2.72	5.22	4.00	5.51	4.59
Tegafur	LPGs	84.33	50.92	65.58	32.57	80.11	47.82	84.57	54.94
Tegafur, Gimeracil and Oteracil Porassium Capsules	LPGs	2.25	1.76	1.75	1.12	2.14	1.65	2.26	1.90
Trastuzumab	OBs	119.01	93.01	92.54	59.48	113.05	87.33	119.34	100.35
Vindesine	LPGs	61.78	46.25	48.04	29.57	58.69	43.43	61.95	49.90
Vinorelbine	OBs	112.93	75.03	87.81	47.98	107.28	70.45	113.24	80.96
	LPGs	69.92	47.71	54.37	30.51	66.42	44.80	70.12	51.47

**Table 8 ijerph-16-03728-t008:** Affordability of anticancer medicines in the urban and rural area in different regions of the Jiangsu Province after reimbursement.

Drug Name	Type	Total (%)		High Income (%)		Middle Income (%)		Low Income (%)	
		2012	2016	2012	2016	2012	2016	2012	2016
**Urban area**									
Calcium folinate	LPGs	5.71	3.45	4.69	2.74	6.09	3.49	7.77	4.51
Capecitabine	LPGs	1.70	0.98	1.40	0.78	1.81	0.99	2.32	1.28
Carboplatin	OBs	0.00	0.00	0.00	0.00	0.00	0.00	0.00	0.00
Carmofur	LPGs	3.67	2.12	3.02	1.68	3.92	2.14	5.00	2.76
Cytarabine	OBs	0.00	0.00	0.00	0.00	0.00	0.00	0.00	0.00
	LPGs	0.00	0.00	0.00	0.00	0.00	0.00	0.00	0.00
Docetaxel	OBs	67.87	32.94	55.82	26.14	72.41	33.28	92.39	43.05
	LPGs	21.31	9.77	17.52	7.76	22.73	9.87	29.00	12.77
Doxorubicin	LPGs	0.00	0.00	0.00	0.00	0.00	0.00	0.00	0.00
Endostar	LPGs	255.50	154.66	210.12	122.75	272.57	156.28	347.78	202.13
Epirubicin	LPGs	7.68	4.59	6.32	3.64	8.19	4.64	10.45	6.00
Erlotinib	OBs	278.16	168.38	228.76	133.64	296.75	170.14	378.63	220.06
Etoposide	LPGs	0.00	0.00	0.00	0.00	0.00	0.00	0.00	0.00
Fluorouracil	LPGs	0.00	0.00	0.00	0.00	0.00	0.00	0.00	0.00
Gefitinib	OBs	58.99	35.71	48.51	28.34	62.93	36.08	80.29	46.67
Gemcitabine	OBs	47.56	27.80	39.11	22.06	50.74	28.09	64.73	36.33
	LPGs	27.68	15.08	22.77	11.97	29.53	15.24	37.68	19.71
Ifosfamide	LPGs	8.41	3.98	6.91	3.16	8.97	4.02	11.44	5.20
Imatinib	OBs	92.07	55.73	75.71	44.23	98.22	56.31	125.32	72.84
Irinotecan	LPGs	22.46	13.59	18.47	10.79	23.96	13.74	30.57	17.77
Lentinan	LPGs	20.67	11.10	17.00	8.81	22.05	11.22	28.14	14.51
Methotrexate	LPGs	0.00	0.00	0.00	0.00	0.00	0.00	0.00	0.00
Mitomycin	LPGs	0.00	0.00	0.00	0.00	0.00	0.00	0.00	0.00
Nedaplatin	LPGs	10.60	4.94	8.72	3.92	11.31	4.99	14.43	6.45
Ondansetron	LPGs	44.90	27.18	36.92	21.57	47.90	27.46	61.11	35.52
Oxaliplatin	OBs	65.90	39.04	54.19	30.98	70.30	39.44	89.70	51.02
	LPGs	10.16	4.42	8.35	3.51	10.84	4.47	13.83	5.78
Paclitaxel	OBs	0.00	0.00	0.00	0.00	0.00	0.00	0.00	0.00
	LPGs	0.00	0.00	0.00	0.00	0.00	0.00	0.00	0.00
Pemetrexed	OBs	113.26	68.56	93.14	54.41	120.83	69.28	154.17	89.60
	LPGs	49.18	29.77	40.44	23.63	52.46	30.08	66.94	38.90
Semustine	LPGs	0.00	0.00	0.00	0.00	0.00	0.00	0.00	0.00
Tegafur	LPGs	11.98	5.60	9.85	4.45	12.78	5.66	16.30	7.32
Tegafur, Gimeracil and Oteracil Porassium Capsules	LPGs	1.28	0.77	1.05	0.61	1.36	0.78	1.74	1.01
Trastuzumab	OBs	67.60	40.92	55.59	32.48	72.12	41.35	92.02	53.48
Vindesine	LPGs	8.77	5.09	7.21	4.04	9.36	5.14	11.94	6.65
Vinorelbine	OBs	16.04	8.25	13.19	6.55	17.11	8.34	21.83	10.79
	LPGs	9.93	5.25	8.17	4.16	10.59	5.30	13.52	6.86
**Rural area**									
Calcium folinate	LPGs	10.04	7.85	7.81	5.02	9.54	7.37	10.07	8.47
Capecitabine	LPGs	3.59	2.67	2.79	1.71	3.41	2.51	3.60	2.88
Carboplatin	OBs	0.00	0.00	0.00	0.00	0.00	0.00	0.00	0.00
Carmofur	LPGs	7.76	5.77	6.03	3.69	7.37	5.42	7.78	6.23
Cytarabine	OBs	0.00	0.00	0.00	0.00	0.00	0.00	0.00	0.00
	LPGs	0.00	0.00	0.00	0.00	0.00	0.00	0.00	0.00
Docetaxel	OBs	143.38	89.84	111.50	57.45	136.21	84.36	143.78	96.93
	LPGs	45.02	26.65	35.01	17.04	42.76	25.03	45.14	28.76
Doxorubicin	LPGs	0.00	0.00	0.00	0.00	0.00	0.00	0.00	0.00
Endostar	LPGs	449.80	351.54	349.77	224.80	427.29	330.08	451.05	379.28
Epirubicin	LPGs	16.23	12.52	12.62	8.00	15.41	11.75	16.27	13.51
Erlotinib	OBs	489.70	382.72	380.80	244.74	465.20	359.37	491.06	412.93
Etoposide	LPGs	0.00	0.00	0.00	0.00	0.00	0.00	0.00	0.00
Fluorouracil	LPGs	0.00	0.00	0.00	0.00	0.00	0.00	0.00	0.00
Gefitinib	OBs	124.62	97.39	96.90	62.28	118.38	91.45	124.96	105.08
Gemcitabine	OBs	100.47	75.82	78.13	48.48	95.44	71.19	100.75	81.80
	LPGs	58.48	41.13	45.48	26.30	55.56	38.62	58.64	44.38
Ifosfamide	LPGs	17.76	10.85	13.81	6.94	16.87	10.19	17.81	11.71
Imatinib	OBs	194.50	152.01	151.25	97.21	184.77	142.73	195.04	164.01
Irinotecan	LPGs	47.44	37.08	36.89	23.71	45.07	34.81	47.57	40.00
Lentinan	LPGs	36.39	25.23	28.30	16.14	34.57	23.69	36.49	27.23
Methotrexate	LPGs	0.00	0.00	0.00	0.00	0.00	0.00	0.00	0.00
Mitomycin	LPGs	0.00	0.00	0.00	0.00	0.00	0.00	0.00	0.00
Nedaplatin	LPGs	22.39	13.47	17.41	8.61	21.27	12.65	22.45	14.53
Ondansetron	LPGs	79.04	61.77	61.46	39.50	75.08	58.00	79.26	66.65
Oxaliplatin	OBs	139.21	106.47	108.26	68.09	132.25	99.97	139.60	114.87
	LPGs	21.46	12.06	16.69	7.71	20.38	11.32	21.52	13.01
Paclitaxel	OBs	0.00	0.00	0.00	0.00	0.00	0.00	0.00	0.00
	LPGs	0.00	0.00	0.00	0.00	0.00	0.00	0.00	0.00
Pemetrexed	OBs	239.27	187.00	186.06	119.58	227.30	175.59	239.94	201.76
	LPGs	103.89	81.19	80.78	51.92	98.69	76.24	104.18	87.60
Semustine	LPGs	0.00	0.00	0.00	0.00	0.00	0.00	0.00	0.00
Tegafur	LPGs	25.30	15.28	19.67	9.77	24.03	14.34	25.37	16.48
Tegafur, Gimeracil and Oteracil Porassium Capsules	LPGs	2.25	1.76	1.75	1.12	2.14	1.65	2.26	1.90
Trastuzumab	OBs	119.01	93.01	92.54	59.48	113.05	87.33	119.34	100.35
Vindesine	LPGs	18.53	13.87	14.41	8.87	17.61	13.03	18.58	14.97
Vinorelbine	OBs	33.88	22.51	26.34	14.39	32.18	21.14	33.97	24.29
	LPGs	20.98	14.31	16.31	9.15	19.93	13.44	21.03	15.44

**Table 9 ijerph-16-03728-t009:** Factors associated with the affordability of anticancer medicines.

Factors	B	Wald	OR (95%CI)	*p*
Survey year (reference = 2012)				
2016	0.53	14.72	1.69 (1.29–2.22)	<0.001
Medicine variety (reference = OBs)				
LPGs	2.75	281.67	15.60 (11.32–21.51)	<0.001
Area (reference = rural area)				
Urban area	0.64	21.34	1.89 (1.44–2.48)	<0.001
Income level (reference = low income)				
Middle income	0.12	0.55	1.13 (0.82–1.56)	0.457
High income	0.46	7.51	1.59 (1.14–2.20)	0.006
Covered by BMI or not (reference = not covered)				
Covered by BMI	1.59	79.39	4.92 (3.46–6.98)	<0.001
WHO essential medicine (reference = non-WHO essential medicine)				
WHO essential medicine	1.01	40.84	2.75 (2.02–3.75)	<0.001

Note: Outcome (y)—affordable or not; y = 0, not affordable; y = 1, affordable; BMI, basic medical insurance in the Jiangsu Province.

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
