# Peer review of "Availability, Price and Affordability of Anticancer Medicines: Evidence from Two Cross-Sectional Surveys in the Jiangsu Province, China"

_ijerph, 2019, doi:10.3390/ijerph16193728_

Round 1

Reviewer 1 Report

Overall, the manuscript would be benefit from using a proof-reading service. Some typo and grammatical errors were spotted in the abstract and manuscript.

The authors assessed availability, price and affordability of pharmacotherapy for cancer in public hospitals in Jiangsu Province, China. This kind of manuscript would be more suitable for a journal related to public health.

Some comments are as follows:

1) why only 40 medicines were selected?

2) how did the results look like when compared to the national results?

3) what items were included in the standardised form?

4) the availability, price and affordability of pharmacotherapy for cancer in public hospitals can be influenced by the types of cancers in the region. The biggest limitation of the study is that this study did not differentiate the type of cancers. Therefore, the results were expressed explicitly which makes the findings not novel.

Author Response

Response to Reviewer 1 Comments

Point 1: Overall, the manuscript would be benefit from using a proof-reading service. Some typo and grammatical errors were spotted in the abstract and manuscript.

Response 1: Thank you for the valuable comments on the manuscript. We apologized for the typo and grammatical errors. To improve the quality of the English, we had the manuscript edited by a native English speaker. We had corrected all of the typo and grammatical errors in the abstract and manuscript. (Line 18, 24, 118-119, 161, 244, 249, 255, 280, 294-295, 341, 363, 521, 534, 567, 598)

Point 2: The authors assessed availability, price and affordability of pharmacotherapy for cancer in public hospitals in Jiangsu Province, China. This kind of manuscript would be more suitable for a journal related to public health.

Response 2: We are very appreciated your comments that will be valuable for our research.  According to the official website of International Journal of Environmental Research and Public Health (IJERPH), IJERPH is an interdisciplinary peer-reviewed journal which covers Environmental Sciences and Engineering, Public Health, Environmental Health, Occupational Hygiene, Health Economic and Global Health Research, etc. Public health is one of the areas IJERPH has covered. Just as you mentioned, this kind of manuscript would be more suitable for a journal related to public health. Therefore, the purposes of this article are consistent with the scope of International Journal of Environmental Research and Public Health.

Point 3: why only 40 medicines were selected?

Response 3: According to the requirements of the WHO/HAI methodology, surveyed medicines were selected based on the local patients’ disease burden and needs. In this study, the anticancer medicines were identified based on the local cancer patients’ disease needs, the WHO essential medicine list, the 2012 national essential medicine list (NEML), feedback from several oncologists experts and literature reviews. First, according to the oncologists’ recommendation and local health statistics, the most commonly prescribed anticancer medicines were selected, which were used for treating five malignancies (lung cancer, gastric cancer, esophageal cancer, liver cancer, and colorectal cancer) with the higher morbidity and fatality in Jiangsu Province. Therefore, although only 40 medicines were selected, the sampled medicines were representative of the anticancer medicines for clinical use in patients with cancer in Jiangsu Province. Second, of the 40 medicines surveyed, 12 were on the WHO essential medicine list and 10 were on the national essential medicine list (NEML). Totally, there were 46 anticancer medicines on the WHO essential medicine list and NEML. The sampled 18 essential medicines covered nearly 40% of the anticancer medicines which were on the WHO essential medicine list and NEML. Therefore, we selected these 40 anticancer medicines as target objects. We have added this description in Line - for short. (Line 152-165)

Point 4: how did the results look like when compared to the national results?

Response 4: After literature search, we could not find the similar studies on the national findings of access to the anticancer medicines in China. Only one study focused on the bidding prices of anticancer medicines in 25 provinces. The main findings of this study are as follows: The bidding price of same category from different manufacturers is dispersed. The prices of originator brands drugs were higher than the generics drugs [53]. Therefore, compared with the national results of prices of anticancer medicines, the main findings of this study were similar. We have added the results of prices comparison in the section of Discussion as follows: This finding was consistent with a study done on the prices of the anticancer medicines in 25 provinces. (Line 483-484)

Refercence

[53] Yuan, J.; Zhou J.; Chen Y.; Liu, Y.; Mao, Z. Analysis of Bidding Price of Anti-tumor Drugs in 25 Provinces (Region, City). China Pharmacy. 2016, 27, 4336-4339. (In Chinese)

Point 5: what items were included in the standardised form?

Response 5: All the medicines surveyed were noted on a pre-designed perform sheet. We rephrased the part of this paragraph as “A standardized data collection form was designed, which made it convenient and efficient for the well-trained research assistants (RAs) to collect information. The items were included in the standardised form as follows: Facility information (Hospital name, Hospital level, Survey date); Medicine information (The medicine is stock in the hospital on the day of data collection or not); Dosage, Strength, Medicine type (OBs/LPGs), Retail price”. (Line 178-182)

Point 6: the availability, price and affordability of pharmacotherapy for cancer in public hospitals can be influenced by the types of cancers in the region. The biggest limitation of the study is that this study did not differentiate the type of cancers. Therefore, the results were expressed explicitly which makes the findings not novel.

Response 6: Thank you so much for the valuable comments on the paper. In order to differentiate the type of cancers and express explicitly, we have added the information on the main indication of each sampled anticancer medicines in Table 2. On the other hand, we have added this description as follows: Of the 40 medicines surveyed, 15 were for the treatment of lung cancer, 12 were for the treatment of gastric cancer, 8 were the treatment of esophageal cancer, 7 were the treatment of liver cancer and 9 were the treatment of colorectal cancer. (Line 165-171)

Reviewer 2 Report

I commend the authors for their comprehensive review on availability, price and affordability for 40 anticancer medications in Jiangsu Province, China.   Since this study is a comparison of three policy outcomes (availability, price and affordability) in two timeframes (2012 vs. 2016), a clearer definition of terms and the impact of the public policy changes should be addressed.  Please see my comments below:

Three medical insurance models in China cover the patients living in the urban or rural region as well as by their employment status. All three models combined may covered 95% of the population.    The medication price determination may be different among those insurance models.   Is it reasonable just using a simple formula (line 189-190) considering the differences in pricing schemes across different insurance models and levels of hospital? I think that affordability should be determined by a patient’s out of pocket expenditures instead of the medication treatment cost per month.  (lines 194-195) Line 170: It states that “Absent: 0% of hospitals: these anticancer medicines were not available in any hospital surveyed”;  I assume the selected medications will be available at least at one of the study hospitals.   Do you need this category? Line 143: Do all 40 medications in the study have both a brand name drug and its generic equivalent entities?   What do you mean by “commonly” used medication?   Line 147: I assume only those brand name medications with a generic equivalent were included in the study and brand name drugs without a generic equivalent were not included in the study.  This practice may introduce selection bias.  Lines 143-145: “A total of 40 medicines were surveyed in both years, which were selected from the commonly used medicines for five malignancies (lung cancer, gastric cancer, esophageal cancer, liver cancer, and colorectal cancer) with the highest fatality rate in Jiangsu Province.” Consequently, new medications that were introduced between 2013 and 2016 are not included.   Will that introduce additional bias in the study on top of the bias introduced lines 426-428? Along the same argument, the price comparison in table 6 is an apple vs orange comparison because you compared a different set of drugs in two study points, 2012 and 2016. New drugs introduced after 2012 were not included in the price calculation.  I expect most of new drugs will carry higher price tag. Please give a clear definition for Lowest Priced Generics (LPGs) in the manuscript. I am not sure the study hospitals will carry more than one generic equivalent medications for a brand name drug in its inventory.  The survey method based on a one day survey may not identify all the generic equivalents in that hospital especially for secondary general hospitals which the authors recognized in the study limitations.(line 426). Please explain why “Types of Hospital” (cancer, tertiary and secondary) was not included in the affordability logistic regression model? Will it have a multicollinearity relationship with the “Area” variable?    Lines 106, 125-126 and 335: What do you mean by “improved” in this sentence “we ‘improved the WHO and HAI standard methodology to estimate the…?” Please discuss how many new anticancer drugs are available in 2016? In general, we should have observed price increase from 2012 to 2016.  Were any price control measures on pharmaceutical products at the industry level implemented between 2012 and 2016? In fact, Table 6 tells a different story.  Was it due to the implementation of the zero-markup policy eliminating 15% markup in the patient price? Line 432: The authors should expand the effect of the zero-markup policy on price decrease at the patient level in 2016 in Table 6 as well as cost shifting due to the zero-markup policy on overall patient affordability.  It has been reported that other fees related to medical examination has been rising.   The authors should explore the reasons for low availability and low affordability of medications in Figures 1, 2, 3, and 4, and suggest the policies to address these two access disparities beyond the discussion in lines 371-375. Tables 3 and 7: Based on the data collection methodology, in addition to %, n  should be included in the table.  It will allows the reader to see the real difference of individual medication availability between two study years.    

Author Response

Response to Reviewer 2 Comments

Point 1: I commend the authors for their comprehensive review on availability, price and affordability for 40 anticancer medications in Jiangsu Province, China. Since this study is a comparison of three policy outcomes (availability, price and affordability) in two timeframes (2012 vs. 2016), a clearer definition of terms and the impact of the public policy changes should be addressed.

Response 1: Thank you for the valuable comments on the paper. We accept your useful suggestions. We have given a clearer definition of terms such as Lowest Priced Generics (LPGs). (Line 114-116). We also analyzed the impact of the public policy changes, such as the effects of  zero-markup policy on price decrease. (Line 549-556)

Point 2: Three medical insurance models in China cover the patients living in the urban or rural region as well as by their employment status. All three models combined may covered 95% of the population. The medication price determination may be different among those insurance models. Is it reasonable just using a simple formula (Line 189-190) considering the differences in pricing schemes across different insurance models and levels of hospital? I think that affordability should be determined by a patient’s out of pocket expenditures instead of the medication treatment cost per month. (Line 194-195)

Response 2: We decided to accept your valuable suggestions. We used a simple formula to measure the differences in pricing schemes across different insurance models and levels of hospital. As your opinion, it is not reasonable. We absolutely agree with you. The affordability should be determined by a patient’s out of pocket expenditures instead of the medication treatment cost per month. In order to overcome this issue, we have calculated the affordability of the surveyed anticancer medicines after health insurance reimbursement. We have added the findings in Table 8 (Line 347-348) and some descriptions in the section of Results as follows: Universal health insurance system has been established in China since 2008, which mainly included basic medical insurance scheme for urban employees, urban residents and rural residents. Currently, both schemes cover the same anticancer medicines divided into two categories referred to as ‘class A’ and ‘class B’. Class A medicines require 0% out-of -pocket payment by patients. Class B medicines require urban employees to pay 25% and general residents to pay 30% of medicine expenditure. The affordability of each medicine with different levels of insurance reimbursement is provided in Table 8. (Line 333-340). After health insurance reimbursement, the affordability were improved a lot. In both years, the proportion of affordable OBs increased to 33.33% and 25% for urban and rural residents respectively. Furthermore, in 2012, the proportion of affordable LPGs increased by 38.46% and 57.69% for urban and rural residents respectively; and in 2016, the proportion of affordable LPGs increased by 23.07% and 42.31% for urban and rural residents respectively. (Line 341-346)

Point 3: Line 170: It states that “Absent: 0% of hospitals: these anticancer medicines were not available in any hospital surveyed”; I assume the selected medications will be available at least at one of the study hospitals. Do you need this category?

Response 3: Yes, we agree with you. In fact, the availability was accessed at each facility only for the anticancer medicines were in stock on the day of data collection, which may not accurately capture the availability of anticancer medicines in these hospitals over time.We have described this issue in the section of Limitation. (Line 557-559)

Point 4: Line 143: Do all 40 medications in the study have both a brand name drug and its generic equivalent entities? What do you mean by “commonly” used medication? Line 147: I assume only those brand name medications with a generic equivalent were included in the study and brand name drugs without a generic equivalent were not included in the study. This practice may introduce selection bias.

Response 4: According to the the official website of National Medical Products Administration of China (NMPA), 33 medications have both a brand name drug and its generic equivalent entities in 2012. Meanwhile, 35 medications have both a brand name drug and its generic equivalent entities In 2016. Therefore, 7 brand name drugs without a generic equivalent were also included in the survey in 2012 and 5 brand name drugs without a generic equivalent were also included in the survey. (Line 171, Table 2)

     In this study, the“commonly” used medication means that the medicines that could satisfy local cancer patients’ disease needs. According to local health statistics of Jiangsu Province, the 5 cancers with the higher morbidity and fatality are as follows: lung cancer, gastric cancer, esophageal cancer, liver cancer, and colorectal cancer. Of the 40 medicines surveyed, 15 were for the treatment of lung cancer, 12 were for the treatment of gastric cancer, 8 were the treatment of esophageal cancer, 7 were the treatment of liver cancer and 9 were the treatment of colorectal cancer. As a result, the sampled medicines were regards as“commonly” used medication in this study. (Line 165-167)

Point 5: Lines 143-145: “A total of 40 medicines were surveyed in both years, which were selected from the commonly used medicines for five malignancies (lung cancer, gastric cancer, esophageal cancer, liver cancer, and colorectal cancer) with the highest fatality rate in Jiangsu Province.” Consequently, new medications that were introduced between 2013 and 2016 are not included. Will that introduce additional bias in the study on top of the bias introduced lines 426-428? Along the same argument, the price comparison in table 6 is an apple vs orange comparison because you compared a different set of drugs in two study points, 2012 and 2016. New drugs introduced after 2012 were not included in the price calculation. I expect most of new drugs will carry higher price tag.

Response 5: Thank you so much for the valuable comments on the paper. We agree with you. Yes, we have added the selection bias in the section of Limitation as follows: The new anticancer medications that were introduced between 2013 and 2016 were not included, which could led to selection bias in the price comparison between 2012 and 2016. Due to most of new drugs with higher price tag, lack of new drugs in 2016 could lead to biased lower prices in 2016. (Line 566-569)

However, we researched for the information on introduction time of the new anticancer medications in China from NMPA. We found that only five new anticancer medicines introduced between 2012 and 2016. Most of new anticancer medicines were introduced after 2016 in China. (Table 1). Therefore, the selection bias could not have great influence on the main findings of price comparison.

            Table 1. The new anticancer medicines between 2013 and 2019 in China

Drug name

manufacturer

Approved time in China

Crizotinib

Pfizer

January, 2013

Lenalidomide

Celgene

January, 2013

Everolimus

Novartis

January, 2013

Apatinib

Hengrui

October, 2014

Axitinib

Pfizer

April, 2015

Afatinib

Boehringer-Ingelheim

March, 2017

Osimertinib Mesylate

AstraZeneca

May, 2017

Pazopanib

Novartis

May, 2017

Regorafenib

Bayer

May, 2017

Ruxolitinib Phosphate

Novartis

August, 2017

Vemurafenib

Roche

October, 2017

Azacitidine

Beigene

February, 2018

Ceritinib

Novartis

May, 2018

Ixazomib Citrate

Takeda

May, 2018

Anlotinib

Chiatai Tianqing

May, 2018

Nivolumab

Bristol-Myers Squibb

June, 2018

Palbociclib

Pfizer

July, 2018

Alectinib

Roche

August, 2018

Pyrotinib Maleate

Hengrui

August, 2018

Olaparib

AstraZeneca

August, 2018

Pembrolizumab

MSD

August, 2018

Lenvatinib Mesilate

Eisai

September, 2018

Fruquintinib

Hutchison MediPharma

September, 2018

Toripalimab

Top Alliance

December, 2018

Sintilimab

Innovent

December, 2018

Pertuzumab

Roche

December, 2018

Camrelizumab

Hengrui

May, 2019

Dacomitinib

Pfizer

July, 2019

Degarelix

Ferring

July, 2019

Point 6: Please give a clear definition for Lowest Priced Generics (LPGs) in the manuscript. I am not sure the study hospitals will carry more than one generic equivalent medications for a brand name drug in its inventory.

Response 6: The OB product had a unique originator pharmaceutical company, and Lowest Priced Generics (LPGs) were defined as the same product sold under the generic name with the lowest unit price at each medicine outlet at the time of data collection. We have added the description in the section of Methods as follows: The OB product had a unique originator pharmaceutical company, and LPGs were defined as the same product sold under the generic name with the lowest unit price at each hospital at the time of data collection. (Line 114-116)

According to the policy issued from the National Health Commission of China, the Chinese hospital can carry more than one generic equivalent medications for a brand name drug in its inventory.

Point 7: The survey method based on a one day survey may not identify all the generic equivalents in that hospital especially for secondary general hospitals which the authors recognized in the study limitations.(line 426). Please explain why “Types of Hospital” (cancer, tertiary and secondary) was not included in the affordability logistic regression model? Will it have a multicollinearity relationship with the “Area” variable?

Response 7: We decided to accept your valuable suggestions. As independent variable, types of Hospital was included in the affordability logistic regression model. However, there’s no significant difference in different hospital types (p=1.00). Collinearity test indicated that there’s no multicollinearity relationship between types of Hospital variable and Area variable (VIF range from 1.000 to 1.314). (Line 407-408, 412-414)

     Due to the bidding system, the pharmaceutical companies that win bids offer the tender prices. As a result, there is relatively little difference between the prices of the same medicine with same strength and dosage form in different types of hospital. Therefore, “Types of Hospital”  failed to enter the logistic regression equation to forecast the affordability of medicines. We have added the explanation in the section of Discussion. (Line 526-530)

Point 8: Lines 106, 125-126 and 335: What do you mean by “improved” in this sentence “we ‘improved the WHO and HAI standard methodology to estimate the…?” Please discuss how many new anticancer drugs are available in 2016?

Response 8: According to the WHO/HAI methodology, generally, if the cost of a course treatment of a medicine is no more than the lowest wage of 1 day, it is considered affordable. A study published by Rasha Khatib et al. [33] defined it as; “If a treatment cost per month less than 20% of the household capacity-to-pay then it can be considered as affordable.” And the method has been performed in two studies of anticancer medicines affordability in Pakistan [39-40]. In this study, this concept modified and affordability was measured for each medicine by low, middle, and high income class of patients in urban and rural areas through this formula. (Line 215-222)

     We apologize that we use incorrect expressions in this sentence. In fact, we did not  improve the WHO and HAI standard methodology. Therefore, we changed the“improved” into “adopted”.(Line 110, 435). We researched for the information on introduction time of the new anticancer medications in China from NMPA. Only five new anticancer medicines introduced between 2012 and 2016. Most of new anticancer medicines were introduced after 2016 in China.

Point 9: In general, we should have observed price increase from 2012 to 2016. Were any price control measures on pharmaceutical products at the industry level implemented between 2012 and 2016? In fact, Table 6 tells a different story. Was it due to the implementation of the zero-markup policy eliminating 15% markup in the patient price?

Response 9: The price control measures on pharmaceutical products at the industry level were not implemented between 2012 and 2016. On the contrary, drug price regulation was replaced by market pricing policy since 2015. Therefore, we can conclude that the observed price decrease from 2012 to 2016 was due to the implementation of the zero-markup policy eliminating 15% markup in the patient price. We have added this opinion in the section of Discussion. (Line 494-500)

Point 10: Line 432: The authors should expand the effect of the zero-markup policy on price decrease at the patient level in 2016 in Table 6 as well as cost shifting due to the zero-markup policy on overall patient affordability. It has been reported that other fees related to medical examination has been rising.

Response 10: We decided to accept your valuable suggestions. We have expanded the effect of the zero-markup policy on price decrease as well as cost shifting as follows: On one hand, due to the implementation of zero-markup policy, drug markups was basically eliminated. It led to price decrease from 2012 to 2016 at the patient level. On the other hand, medical cost was shifted from medicines to other medical service such as surgery and outpatient consultation. For instance, some empirical studies in Jiangsu Province demonstrated that the total cost of medical care has increased by about 10% with the decline of drug expense after the implementation of zero-markup policy. Meanwhile, the outpatient consultation and treatment fees have increased significantly[60]. The individual patient out-of-pocket expense has decreased slightly [61-62]. (Line 549-556)

References

[60]Huang W, Chen D, Zhao G. Reinforcing the Coonprehensive Pricing and Compensation

Reform of Urban Public Hospital: case of Yangzhou. Chinese Journal of Health Economics,2016(10):58-60. (in Chinese)

[61] Cao Y, Chen J. Thoughts of Stratigies of Public Hospital Medical Pricing Reform: a

Survey of Urban Public Hospitals in Jiangsu. Modern Commercial and Trade Industry,2017(14):130-132. (in Chinese)

[62] Hu D, Xu J, Zhang Y. Impact of removing drug markups on medical insurance fund: practice from Jiangsu. Chinese Journal of Medical Insurance,2017(07):20-24. (in Chinese)

Point 11: The authors should explore the reasons for low availability and low affordability of medications in Figures 1, 2, 3, and 4, and suggest the policies to address these two access disparities beyond the discussion in lines 371-375.

Response 11: Thank you so much for the valuable comments on the paper. We have given the reasons for low availability of medications as follows: The low availability of anticancer medicines may be likely multifactorial, such as inadequate investment in research and development[38], insufficient incentive on maintaining stocks and inefficient procurement systems[47]. (Line 454-456). The reasons for low affordability of medications are as follows: The costly anticancer medicines are protected by product patent rights, which make them expensive monopolies[55]. Generally, the patent of new medicines lasts 20 years, which aims to promote the pharmaceutical firm to recover the costs of research and development and invest again in the new research[56]. In addition, high taxation contributes to the sky-high price of these OBs, which may be a cruel joke for the cancer sufferers[39]. (Line 503-508)

     We have put forward suggestions on the policies to address these two access disparities in the section of Discussion as follows: Policy should focus on addressing the inequity in anticancer medicines affordability between patients with different income levels, and between those in urban and rural areas [35]. Our study also revealed generic substitution could lighten the cost of anticancer medicines to cancer suffers. Governments should adjust related policies and regulations to make the approvals of generic anticancer medicines faster and improve their access to market [59]. The preferential taxation policy and other incentive measures can be given the non-profit generic producers in order to encourage them to manufacture low-cost high-quality anticancer medicines [51]. The government should ensure that first-line LPGs are dispensed in all the public hospitals which can provide services to cancer suffers, and are available for patients and preferentially prescribed [47]. (Line 540-548).  In the section of Conclusion: First, having acknowledged the existence of inequities in availability and affordability, a rise in the public expenditure on cancer is indispensable. In consideration of the low availability of anticancer medicines in Jiangsu Province, priority attention is required for availability in hospitals which can provide cancer services. Second, governments should consider using their bargaining power to reduce prices in purchase phase and abolish taxes on anticancer medicines. Third, policy should focus on the special health insurance plan for low income population of cancer suffers and lowering out-of-pocket payments for cancer therapy. Finally, the health authorities should adjust related policies to make the approvals of generic anticancer medicines faster and improve their access to market. The goal of drug policy is to ensure that first-line generic drugs are dispensed in public sectors and are available for cancer patients and preferentially prescribed. (Line 578-588)

Point 12: Tables 3 and 7: Based on the data collection methodology, in addition to %, n should be included in the table. It will allows the reader to see the real difference of individual medication availability between two study years.

Response 12: We have added “n” in the Table 3. (Line 264). In this study, the affordability of anticancer medicines was assessed by comparing the total cost of medicines per month to the average household monthly income. In Table 7, the “%” refered to the ratio of cost of anticancer medicines per month to the average household monthly income. We could not calculate the numbers in this case. As a result, we could not add “n” in the Table 7.

Reviewer 3 Report

Comments

The paper uses data from one providence in China. What is the external validity of the findings that are presented in the paper? Does China have universal health care system? How prices of medicines are regulated in China? The concluding section of the paper should discuss more about the practical policy lessons that can be drawn from the results.

Author Response

Response to Reviewer 3 Comments

Point 1: The paper uses data from one providence in China. What is the external validity of the findings that are presented in the paper?

Response 1: The external validity of the findings that are presented in the paper is as follows:  Jiangsu Province is located in the eastern region of China, which is in the middle-upper level of development of China. As the first anticancer medicine survey to apply the WHO/HAI methodology to the eastern region of China, the findings of this study provide a comprehensive report on availability, prices and affordability of anticancer medicines in East China. However, due to differences in economic development between Jiangsu and other provinces, there may be regional differences in the affordability of anticancer medicines. We have added the external validity of the findings in the section of Discussion. (Line 439-445)

Point 2: Does China have universal health care system?

Response 2: Yes, China have universal health insurance system. We have added this description as follows: Universal health insurance system has been established in China since 2008, which mainly included basic medical insurance scheme for urban employees, urban residents and rural residents. Currently, both schemes cover the same anticancer medicines divided into two categories referred to as ‘class A’ and ‘class B’. Class A medicines require 0% out-of -pocket payment by patients. Class B medicines require urban employees to pay 25% and general residents to pay 30% of medicine expenditure. (Line 333-340)

Point 3: How prices of medicines are regulated in China?

Response 3: In China, before June 2015, the prices of medicines were regulated by the government. However, it was reported that price regulations had no effect on pharmaceutical price in China[54]. Drug price regulation was replaced by market pricing policy since June 2015. In this study, we compared changes in the prices of different anticancer medicine groups between 2012 and 2016. A price increase from 2012 to 2016 was not observed in this study. Therefore, the market pricing policy could not impose negative impact on prices of anticancer medicines. We have added this opinion in the section of Discussion. (Line 494-500)

References

[54]Meng QY, Cheng G, Silver L, Sun XJ, Rehnberg C, Tomson G. The impact of China's retail drug price control policy on hospital expenditures: a case study in two Shandong hospitals. Health Policy Plan. 2005;20(3):185‐196.

Point 4: The concluding section of the paper should discuss more about the practical policy lessons that can be drawn from the results.

Response 4: Considering the key role of generic medicines in accessibility of anticancer medicines, we have added more details about the practical policy lessons that can be drawn from the results in the section of Conclusion as follows: First, having acknowledged the existence of inequities in availability and affordability, a rise in the public expenditure on cancer is indispensable. In consideration of the low availability of anticancer medicines in Jiangsu Province, priority attention is required for availability in hospitals which can provide cancer services. Second, governments should consider using their bargaining power to reduce prices in purchase phase and abolish taxes on anticancer medicines. Third, policy should focus on the special health insurance plan for low income population of cancer suffers and lowering out-of-pocket payments for cancer therapy. Finally, the health authorities should adjust related policies to make the approvals of generic anticancer medicines faster and improve their access to market. The goal of drug policy is to ensure that first-line generic drugs are dispensed in public sectors and are available for cancer patients and preferentially prescribed. (Line 578-588)

Round 2

Reviewer 1 Report

I have no further comments. But the manuscript can still be improved after some minor English and formatting errors are corrected.